# Forecasting Food Inflation in Real Time with Tabular Foundation Models

**Mason Linsky** [1]

## Abstract

Food-price inflation is volatile and difficult to forecast, especially around structural breaks where historical relationships cease to hold. We build a monthly forecasting pipeline using public data from FRED to predict U.S. Food CPI dynamics from 1967–2025, evaluating models with strict time-based splits and publication-lag rules to reduce look-ahead bias. On a challenging post-shock test window (Dec. 2023–Dec. 2025), TabPFN is the only method with positive out-of-sample skill (MAE 1.16; $R^2$ 0.69), while all other models yield negative $R^2$ values. Feature-importance diagnostics reveal that failed models rely on temporal trend extrapolation rather than stable economic covariates, consistent with a regime shift. These results provide evidence that tabular foundation models can be unusually robust on small, low-frequency macro-financial datasets where standard ML pipelines break down.

## 1. Introduction

Food price inflation matters because grocery spending is frequent and difficult to postpone, representing roughly 10–15% of household income (USDA ERS, 2025). When prices rise, households adjust immediately: switching brands, buying different cuts of meat, or leaning more on staples. Short-horizon forecasting of food CPI is therefore relevant for planners, researchers, and macro-financial decision-makers.

Forecasting inflation is notoriously difficult, especially around turning points. Classic results show that simple benchmarks can be hard to beat when the inflation process changes (Atkeson & Ohanian, 2001; Faust & Wright, 2013; Stock & Watson, 2007). Food CPI compounds this difficulty because its drivers span heterogeneous channels:

energy and transportation costs, labor and demand conditions, trade pressures, climate stress, and global commodity disruptions can each dominate in different periods (FAO, n.d.). Predictive relationships estimated in one regime may not transfer to another.

Foundation-model ideas are increasingly influencing finance and economics, but most work has focused on text-based LLMs (Wu et al., 2023; Yang et al., 2023) or time-series foundation models (Ansari et al., 2024; Das et al., 2023). Tabular foundation models—pre-trained on synthetic tasks and applied via in-context learning—have only recently become practical (Hollmann et al., 2025), and evidence on their behavior in macro-financial settings remains limited.

This paper addresses that gap. We compare TabPFN against five standard tabular ML methods on a long historical sample with a post-2022 regime shift, under a realistic evaluation protocol. Our contributions are: (1) a real-time evaluation pipeline with publication-lag rules for food CPI forecasting; (2) the first empirical evidence, to our knowledge, of tabular foundation model robustness under a macroeconomic regime shift; and (3) failure-mode diagnostics showing why standard models collapse.

## 2. Related Work

**Inflation forecasting and structural instability.** Traditional inflation forecasting begins with univariate methods such as autoregressions and ARIMA (Box & Jenkins, 1970). A key lesson from decades of evidence is that simple benchmarks often perform competitively, particularly when multivariate relationships are unstable (Atkeson & Ohanian, 2001; Faust & Wright, 2013). Inflation dynamics can change over time, making previously useful predictors unreliable; in econometrics, these changes are framed as structural breaks or regime shifts (Bai & Perron, 2003; Stock & Watson, 2007).

**Real-time evaluation.** A recurring concern in macro forecasting is that "final" data are not what was available in real time. Vintage archives such as ALFRED exist precisely because revisions and release timing can materially affect evaluation (Croushore & Stark, 2001; Alfred, 2026). Nowcasting methods emphasize continuously updating pre-

[1]Pine Crest School, Fort Lauderdale, Florida, USA. Correspondence to: Mason Linsky <mason.linsky@pinecrest.edu>.

*Proceedings of the 43rd International Conference on Machine Learning*, Seoul, South Korea. PMLR 306, 2026. Copyright 2026 by the author(s).

dictions as new releases arrive and explicitly accounting for information flow (Giannone et al., 2008). These practices motivate the publication-lag rules used in this paper.

**Tabular ML and foundation models.** For tabular tasks, gradient-boosted trees remain strong baselines (Chen & Guestrin, 2016; Ke et al., 2017). A large empirical literature finds that deep tabular architectures do not consistently outperform boosted trees without careful tuning (Grinsztajn et al., 2022; Shwartz-Ziv & Armon, 2022). FT-Transformer adapts transformer attention to tabular features by tokenizing columns (Gorishniy et al., 2021). TabPFN takes a different route: it is pre-trained across many synthetic tabular tasks and applied to new datasets via in-context learning (Hollmann et al., 2022; 2025). The central hypothesis tested here is whether this pre-training yields robustness benefits in macro-financial forecasting settings with limited samples and potential regime change.

## 3. Data and Methods

### 3.1. Target and Features

The target is a monthly food price index (food-related CPI) from FRED (FRED, 2026), spanning January 1967 through December 2025 (708 monthly timestamps). We collect public indicator series covering energy costs (oil and gasoline price proxies), labor market conditions (unemployment rate), consumer sentiment (OECD, 2026), import prices, agricultural commodity proxies, monetary policy (federal funds rate), and financial stress indices. After initial collection, the dataset contained 36 raw columns. Series were standardized to monthly frequency; columns with complete missingness were removed (3 dropped). For intermittent gaps, we forward-filled and added missingness flags.

After cleaning, we engineered 21 additional features: temporal encodings (year, month, quarter, cyclical sine/cosine transforms), lag features (1, 3, 6, and 12-month lags of the target), rolling statistics (3, 6, and 12-month rolling means and standard deviations computed with `shift(1)` to prevent leakage), and linear/squared trend terms. The final feature count is 53.

### 3.2. Evaluation Protocol

To approximate what would have been known at forecast time, we implement conservative publication-lag rules: predictors are shifted so that month-$t$ forecasts do not use contemporaneous values that would plausibly be released after the end of month $t$ (Croushore & Stark, 2001; Alfred, 2026). We use strict time-based splits with no shuffling:

- **Training:** Jan. 1967 – Nov. 2022
- **Validation:** Dec. 2022 – Nov. 2023

*Table 1.* Test set performance (Dec. 2023–Dec. 2025). Lower is better for MAE, RMSE, MAPE, and Max Error; higher is better for $R^2$.

| Model | MAE | RMSE | $R^2$ | MAPE | Max Err. |
|---|---|---|---|---|---|
| TabPFN | 1.16 | 2.93 | 0.69 | 0.34 | 13.97 |
| FT-Trans. | 26.37 | 32.76 | $-38.41$ | 7.92 | 117.61 |
| Neural Net | 26.56 | 36.08 | $-46.82$ | 7.98 | 87.97 |
| XGBoost | 29.84 | 31.02 | $-34.35$ | 8.97 | 63.44 |
| Rand. Forest | 33.94 | 34.85 | $-43.61$ | 10.21 | 63.40 |
| LightGBM | 51.03 | 51.57 | $-96.68$ | 15.36 | 72.12 |

- **Test:** Dec. 2023 – Dec. 2025

The test period captures a post-inflation-shock environment where relationships may differ from the long pre-2023 training history.

### 3.3. Models

We train six model families:

**TabPFN** (Hollmann et al., 2022; 2025): pre-trained tabular foundation model, used with default configuration. To respect inference constraints, we remove zero-variance features, select up to the top 100 features by correlation, and cap the context window to $\min(800, N_{\text{train}})$ observations.

**XGBoost** (Chen & Guestrin, 2016): 500 estimators, max depth 6, learning rate 0.01, subsample 0.8, colsample_bytree 0.8, gamma 0.01. **LightGBM** (Ke et al., 2017): 500 estimators, 31 leaves, learning rate 0.01, early stopping patience 50 on validation RMSE. **Random Forest**: 800 trees, unlimited depth, min 1 sample per leaf. **Feedforward NN**: 3 hidden layers (128→64→32), ReLU, 20% dropout, AdamW with MAE loss, early stopping patience 30, gradient clipping at norm 1.0. **FT-Transformer** (Gorishniy et al., 2021): 64-dim tokens, 8 heads, 4 layers, 256-dim FFN, 15% dropout, AdamW, early stopping patience 40.

We also evaluate an equal-weight ensemble of all six model predictions as a robustness check. We report MAE, RMSE, MAPE, Max Error, and $R^2$.

## 4. Results

Table 1 shows a stark divide: TabPFN is the only model that achieves meaningful predictive skill on the post-shock test window (MAE 1.16; $R^2$ 0.69). All other models have negative $R^2$ values, performing worse than a naive baseline that predicts the test-period mean. The magnitude of the failures is not subtle—FT-Transformer and the neural network have MAE $\approx$ 26 CPI points, and LightGBM degrades further (MAE $\approx$ 51). Even the equal-weight ensemble failed (MAE 23.72; $R^2$ $-20.90$), because five of six component models were systematically wrong, overwhelming TabPFN's correct

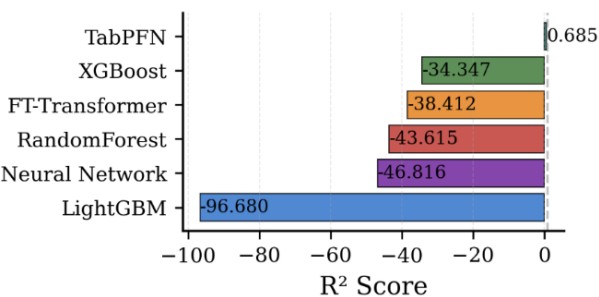

*Figure 1.* $R^2$ scores on the test period by model. Only TabPFN achieves a positive $R^2$; all others produce large negative values indicating worse-than-mean performance.

*Table 2.* Validation vs. test performance. Even TabPFN degrades from validation to test, consistent with distribution shift.

| Model | Val $R^2$ | Test $R^2$ | Val MAE | Test MAE |
|---|---|---|---|---|
| TabPFN | 0.99 | 0.69 | 0.15 | 1.16 |
| FT-Trans. | $-9.81$ | $-38.41$ | 7.19 | 26.37 |
| Neural Net | $-17.44$ | $-46.82$ | 8.79 | 26.56 |
| XGBoost | $-43.75$ | $-34.35$ | 15.20 | 29.84 |
| Rand. Forest | $-43.91$ | $-43.61$ | 15.22 | 33.94 |
| LightGBM | $-187.65$ | $-96.68$ | 31.47 | 51.03 |

signal.

This pattern is consistent with evidence that deep tabular architectures can be fragile on heterogeneous tabular problems unless inductive biases align with the task (Grinsztajn et al., 2022; Shwartz-Ziv & Armon, 2022). Here, even boosted-tree baselines perform poorly, suggesting the primary difficulty is distribution shift rather than model class.

Table 2 shows that even TabPFN experiences a substantial drop from validation ($R^2 \approx 0.99$) to test ($R^2 \approx 0.69$), consistent with a meaningful distribution shift. The other models are already poor on validation and deteriorate further, underscoring that standard hyperparameterized approaches can be unstable when the data-generating process changes between adjacent macroeconomic periods.

Figure 2 reinforces this: failed models often behave like biased or overly smooth extrapolators that do not track the realized 2024–2025 dynamics. Figure 3 further shows that most failed models are not merely noisy; they exhibit large, persistent errors consistent with systematic miscalibration under the new regime.

**Feature importance.** Figure 4 shows that tree-based models assign the highest importance to temporal features (trend, time index, and lagged target values), with macro covariates contributing comparatively little. This is consistent with a class of "trend extrapolators": they appear strong during

stable regimes but fail when the trajectory shifts. The result aligns with broader inflation forecasting evidence that multivariate predictors can be episodically useful and unstable across regimes (Faust & Wright, 2013; Stock & Watson, 2007).

## 5. Discussion

**Why might TabPFN be more robust?** TabPFN is not merely a flexible function approximator trained on one dataset; it is a learned inference procedure pre-trained across many synthetic tasks and applied via in-context learning (Hollmann et al., 2025). This pre-training can act as a strong inductive bias for small datasets, a realistic constraint in monthly macroeconomic forecasting. In contrast, models tuned on a long historical regime may over-specialize to pre-2023 relationships, especially when the strongest in-sample signals come from trend persistence rather than stable causal drivers. TabPFN also requires minimal tuning, whereas the other models underwent hyperparameter optimization that may have inadvertently encouraged overfitting to training-period patterns.

**Regime shift and concept drift.** The results are consistent with the broader concept-drift perspective: the mapping from predictors to outcomes can change over time (Bai & Perron, 2003; Gama et al., 2014). In such settings, the most important question is not whether a model fits historical data, but whether it can detect and gracefully respond to operating outside its training distribution. Practically, this motivates augmenting forecasting systems with (i) drift detection and (ii) uncertainty quantification that can flag low-trust predictions, especially in periods that look unlike the historical training regime. Classic inflation forecasting evidence already emphasizes humility: predictive relationships can weaken and simple benchmarks can be competitive (Atkeson & Ohanian, 2001; Faust & Wright, 2013). Our results extend that lesson to modern tabular ML pipelines under a contemporary, out-of-sample regime.

**Implications for structured-data foundation models.** Foundation-model adoption in finance has largely emphasized unstructured text and time-series models. Many operational finance and economics workflows, however, remain dominated by structured tabular panels (macro indicators, risk factors, accounting features, and market microstructure summaries). This study provides a concrete, reproducible macro-financial case showing that tabular foundation models can offer robustness benefits on small, low-frequency datasets. The finding that TabPFN's in-context learning generalizes better than models explicitly trained on the target dataset suggests that breadth of pre-training experience may be more valuable than depth of task-specific optimization when operating near distribution boundaries.

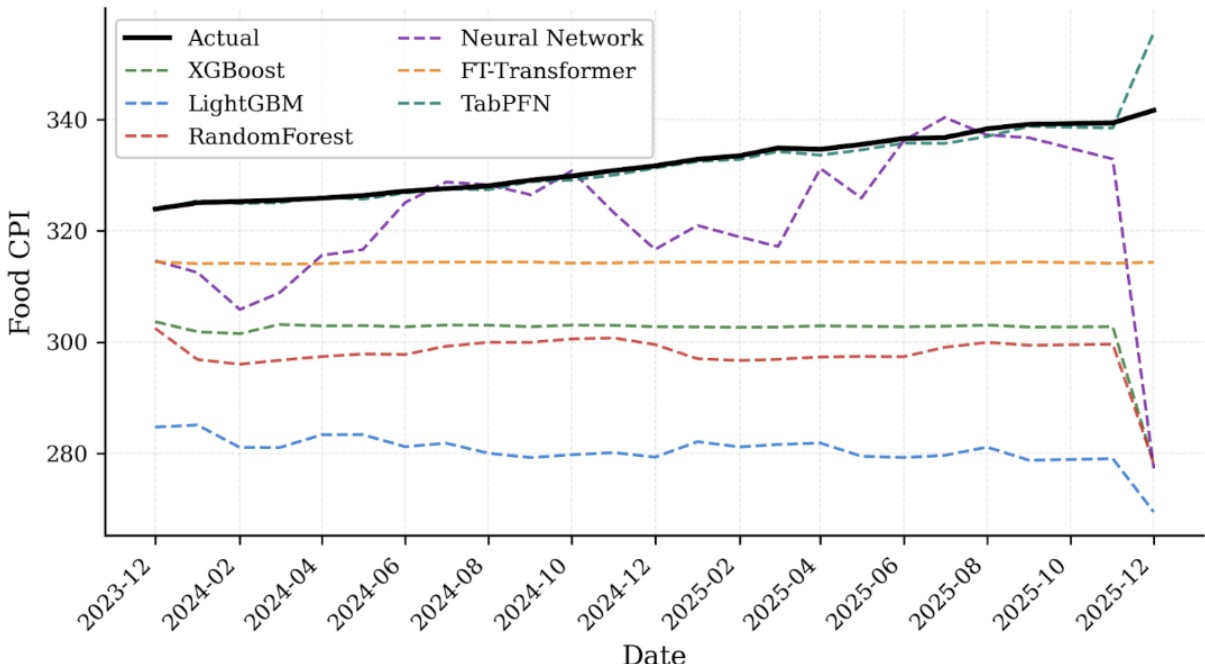

*Figure 2.* Actual vs. predicted Food CPI over the test period (Dec. 2023–Dec. 2025). TabPFN tracks the actual series closely, while other models diverge or flatten, especially toward the end of the period.

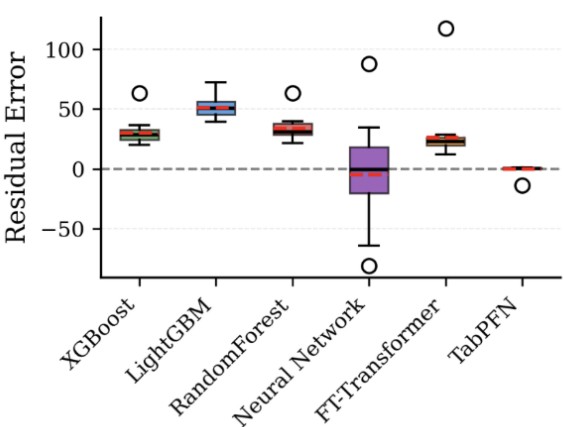

*Figure 3.* Residual error distributions by model on the test period. TabPFN residuals cluster near zero with narrow spread; other models show large dispersion and systematic bias.

## 6. Limitations and Future Work

Aggregation can hide intra-month dynamics and release-calendar effects. The feature set is constrained by public availability and does not include supply-chain or policy-shock measures that could improve responsiveness. While we implement conservative lag rules, fully vintage-correct

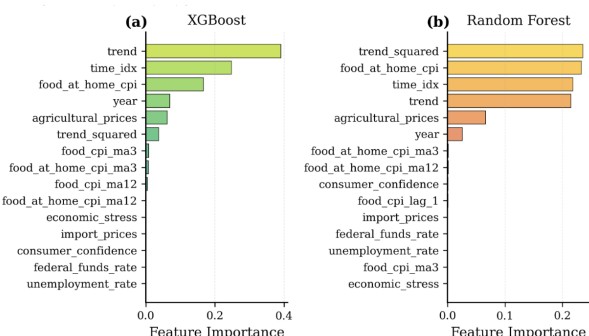

*Figure 4.* Feature importance for tree-based models. (a) XGBoost and (b) Random Forest assign highest importance to time-trend features, with economic indicators contributing much less.

evaluation would require reconstructing exact data vintages via ALFRED (Croushore & Stark, 2001; Alfred, 2026). The tuning protocol may introduce comparison bias: TabPFN uses defaults while other models use fixed hyperparameters with limited tuning; a systematic hyperparameter search with equal budget across all tunable models would strengthen the comparison. Future work should address explicit regime detection and drift monitoring (Bai & Perron, 2003; Gama et al., 2014), uncertainty quantification, online updating to adapt as new regimes emerge, and richer drivers including supply-chain proxies and shipping rates.

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
