# OpenReview forum: "Forecasting Food Inflation in Real Time with Tabular Foundation Models"
_ICML.cc/2026/Workshop/FMSD — FMSD @ ICML 2026 Poster_

### Official Review · Reviewer_u4CE · 2026-05-19
**Promising application, but empirical comparison needs stronger validation**

**Rating:** 5
**Confidence:** 4

**Review:**

### Summary

This paper studies real-time forecasting of U.S. food CPI using public macroeconomic indicators and a monthly forecasting pipeline from 1967–2025. The authors compare TabPFN against several standard tabular ML baselines under strict time-based splits and publication-lag rules. The main result is that TabPFN achieves strong positive out-of-sample performance on the Dec. 2023–Dec. 2025 test period, while all other evaluated models obtain strongly negative $R^2$ scores.

### Strengths

The paper addresses a relevant structured-data forecasting problem that fits the workshop scope well. Food inflation forecasting under regime shift is practically important, and the use of a tabular foundation model for this setting is interesting.

The real-time evaluation setup is a strength. The use of strict time-based splits and publication-lag rules is appropriate for macroeconomic forecasting and helps reduce look-ahead bias .

The dataset and pipeline may be one of the paper’s most valuable contributions. The authors combine a long monthly history with macroeconomic, commodity, labour, sentiment, monetary, and financial-stress indicators, and engineer lagged targets, rolling statistics, and temporal features . If released, this could be useful to the workshop community.

### Areas for Improvement

My main concern is that the empirical result is unusually extreme and needs stronger validation. TabPFN reports MAE 1.16 and $R^2$
=0.69, while all other models have highly negative $R^2$ values and MAEs between 26.37 and 51.03 . This may be possible, but the size of the gap makes the main conclusion difficult to assess without additional checks.

In particular, Figure 2 shows that the non-TabPFN models appear to systematically underforecast the test period, rather than merely producing noisier predictions . This could reflect genuine extrapolation failure, but it could also arise from preprocessing, feature alignment, scaling, inverse transformation, or baseline configuration choices. The paper should rule these possibilities out more explicitly.

### Detailed Comments
- Please strengthen the evidence for the central robustness claim. The paper claims that “tabular foundation models can be unusually robust on small, low-frequency macro-financial datasets where standard ML pipelines break down,” but this is not yet fully supported by the current comparison. TabPFN performs extremely well, while all other baselines perform very poorly, with highly negative $R^2$ values and large MAEs . This gap may be real, but the paper needs to show more clearly that it is not due to baseline configuration or implementation choices. I would recommend adding an appendix with enough baseline detail to verify feature parity, lag construction, preprocessing/scaling, and prediction handling across models. The authors should also include at least a few simple forecasting baselines that would be expected to perform reasonably on a smooth CPI series, such as persistence, seasonal naive, and a linear autoregressive model.
- Please contextualise the very negative $R^2$ values. Since Figure 2 suggests that the non-TabPFN models systematically underforecast the test period rather than simply making noisier predictions, it would help to report the train/test target ranges and compare against mean and persistence predictors. This would make it easier to distinguish genuine extrapolation failure from possible baseline issues.
- Please clarify whether the dataset construction code, processed features, exact splits, and baseline implementations will be released. The dataset/pipeline seems like one of the strongest contributions of the paper, and because the reported result is surprising, reproducibility is especially important.
- The limitations section acknowledges possible comparison bias, but this should be treated more centrally. The paper’s main conclusion depends directly on the fairness and correctness of the baseline comparison.

### Justification of Score

The paper is relevant, clearly written, and studies an interesting application of tabular foundation models to real-world structured forecasting. However, the main empirical claim relies on a very large performance gap between TabPFN and all other baselines. Without simple time-series baselines, fuller baseline documentation, and a clearer reproducibility plan, I am not yet convinced that the evidence fully supports the stronger claims about TabPFN robustness. I would be more positive if the authors added the requested sanity checks and clarified whether the dataset and baseline code will be published.

---

### Official Review · Reviewer_SYWZ · 2026-05-20

**Rating:** 8
**Confidence:** 4

**Review:**

The paper is well-motivated and well-written. I also greatly appreciated the insight from Figure 1 as to how TabPFN is the only model with a meaningful predictive skill on a post-shock test window. This is a great discussion to have as there might be interesting scenarios, such as the so-called "phase changes" where actually the problem is the heavy reliance of the model on trends and seasonality and a weak reliance on covariates.

I would just encourage the readers to rework the last section. It reads too in the weeds with a lot of details whose reference and appearance might seem random. Maybe a discussion of an ablation on the main result of TabPFN could be a good substitute. The ablation that I would suggest would be to take out the covariates (1 by 1, in an intuitive order that makes sense) and understand when does TabPFN also starts to worsen.